# Humic Acid-Coated Fe_3_O_4_ Nanoparticles Confer Resistance to Acremonium Wilt Disease and Improve Physiological and Morphological Attributes of Grain Sorghum

**DOI:** 10.3390/polym14153099

**Published:** 2022-07-30

**Authors:** Sherif Mohamed El-Ganainy, Amal M. El-Bakery, Heba M. Hafez, Ahmed Mahmoud Ismail, Ali Zein El-Abdeen, Abed Abd Elgalel Ata, Omar A. Y. Abd Elraheem, Yousef M. Y. El Kady, Ahlam F. Hamouda, Hossam S. El-Beltagi, Wael F. Shehata, Tarek A. Shalaby, Ahmed Osman Abbas, Mustafa Ibrahim Almaghsla, Muhammad N. Sattar, Zafar Iqbal

**Affiliations:** 1Department of Arid Land Agriculture, College of Agriculture & Food Sciences, King Faisal University, P.O. Box 420, Al-Ahsa 31982, Saudi Arabia; tshalaby@kfu.edu.sa (T.A.S.); malmghaslah@kfu.edu.sa (M.I.A.); 2Vegetable Diseases Research Department, Plant Pathology Research Institute, Agricultural Research Center (ARC), Giza 12619, Egypt; 3Maize and Sugar Crops Diseases Research Department, Plant Pathology Research Institute, Agricultural Research Center (ARC), Giza 12619, Egypt; elbakry_aml@yahoo.com (A.M.E.-B.); ali_zein_55@yahoo.com (A.Z.E.-A.); esraaabed1012004@gmail.com (A.A.E.A.); 4Department, of Sorghum Research, Field Crops Research Institute, Agricultural Research Center (ARC), Giza 12619, Egypt; hobahafez67@yahoo.com (H.M.H.); omar5556666omar@gamil.com (O.A.Y.A.E.); youssefelkady82@gmail.com (Y.M.Y.E.K.); 5Department of Forensic Medicine and Toxicology, Teaching Hospital, Faculty of Veterinary Medicine, Benha University, Benha 13736, Egypt; ahlamhamouda10@gmail.com; 6Department of Agricultural Biotechnology, College of Agricultural and Food Science, King Faisal University, P.O. Box 400, Al-Ahsa 31982, Saudi Arabia; helbeltagi@kfu.edu.sa (H.S.E.-B.); wshehata@kfu.edu.sa (W.F.S.); 7Biochemistry Department, Faculty of Agriculture, Cairo University, Gamma St, Giza 12613, Egypt; 8Department of Animal and Fish Production, College of Agricultural and Food Sciences, King Faisal University, P.O. Box 420, Al-Ahsa 31982, Saudi Arabia; aabbas@kfu.edu.sa; 9Central laboratories, King Faisal University, P.O. Box 420, Al-Ahsa 31982, Saudi Arabia; mnsattar@kfu.edu.sa (M.N.S.); zafar@kfu.edu.sa (Z.I.)

**Keywords:** *Acremonium striticum*, acremonium wilt, humic acid, iron oxide nanoparticles, grain sorghum, toxicity

## Abstract

Acremonium wilt disease affects grain quality and reduces sorghum yield around the globe. The present study aimed to assess the efficacy of humic acid (HA)-coated Fe_3_O_4_ (Fe_3_O_4_/HA) nanoparticles (NPs) in controlling acremonium wilt disease and improving sorghum growth and yields. During the season 2019, twenty-one sorghum genotypes were screened to assess their response to *Acremonium striticum* via artificial infection under field conditions and each genotype was assigned to one of six groups, ranging from highly susceptible to highly resistant. Subsequently, over the two successive seasons 2020 and 2021, three different concentrations of 10, 40 and 80 mg L^−1^ of Fe_3_O_4_/HA NPs were tested against *A. striticum*. The concentrations of 40 and 80 mg L^−1^ were found to be highly effective in controlling acremonium wilt disease on different sorghum genotypes: LG1 (highly susceptible), Giza-3 (susceptible), and Local 119 (resistant) genotypes. After harvest, the physiological (growth and yield) and biochemical (peroxidase, catalase, and gibberellic acid) attributes of sorghum plants were determined, and the results demonstrated that concentrations of 40 and 80 mg L^−1^ increased peroxidase and catalase activities in healthy (uninoculated) sorghum genotypes compared to inoculated sorghum genotypes. Additionally, the toxicity of Fe_3_O_4_/HA NPs on male albino rats was investigated via hematological (CBC), chemical (ALT and AST) and histopathological analyses. The concentration 80 mg L^−1^ of Fe_3_O_4_/HA NPs caused a marked increase in ALT and creatinine level after 51 days of feeding. Severe pathological alterations were also observed in liver and kidney tissues of rats administered with grain sorghums treated with 80 mg L^−1^. In comparison with the untreated control plants, a concentration of 40 mg L^−1^ significantly increased the growth, yield and gibberellic acid levels (*p* ≤ 0.05) and was found to be safe in male albino rats. Conclusively, a concentration of 40 mg L^−1^ of Fe_3_O_4_/HA NPs showed promising results in curtailing *A. striticum* infections in sorghum, indicating its great potential to substitute harmful fertilizers and fungicides as a smart agriculture strategy.

## 1. Introduction

Sorghum (*Sorghum bicolor* L. Moench) is one of the most important cereal crops in Egypt and worldwide. It ranks fifth in production after wheat, rice, maize and barley, with an annual global production of 58.71 million tons from a cultivation area of 40.21 million hectares [1]. Acremonium wilt of Sorghum caused by *Acremonium strictum* Gams (also referred to as *Cephalosporium acremonium* Cord) is a leading cause of yield losses (15–60%) to susceptible sorghum cultivars [2,3]. *A. strictum* directly affects sorghum production by poor grain filling or indirectly through lodging or stalk breakage [4,5,6]. The disease is predominating in upper Egypt [7], where the environmental conditions are favorable for sorghum production. Acremonium wilt is not considered as a stalk rot disease because *A. strictum* behaves like a true vascular soil-borne fungus, while stalk-rotting fungi are frequently found in wilted plants, therefore *A. strictum* acts as a predisposing agent.

Smart agriculture is a contemporary approach to achieve sustainability, both short- and long-term, in the countenance of climate change [8,9]. Therefore, the implementation of nanotechnology research in plant protection and production in the agriculture sector is a critical key factor for sustainable developments [10]. Metal-based nanomaterials (NPs) have a much greater surface area-to-volume ratio and unique antimicrobial effects compared to their normal forms. Different NPs have successfully been used to control different fungal, bacterial and viral plant pathogens [10,11,12,13,14,15,16,17,18]. Iron-oxide-based nanomaterials have become of great interest to scientists and researchers due to their favorable magnetic and electrical properties. They have many interesting applications in industry [19,20,21,22,23], medicine and biology [24,25]. Iron oxide nanomaterials were mainly divided into three types, i.e., α-Fe_2_O_3_, γ-Fe_2_O_3_ and Fe_3_O_4_. Among all, maghemite (γ-Fe_2_O_3_) and magnetite (Fe_3_O_4_) are mostly studied [26,27] and frequently used due to their strong magnetic targeting and structural features [28]. However, Fe_3_O_4_ NPs, stand-alone, have a tendency for agglomeration, due to strong magnetic attraction among particles through van der Waals forces [29]. Therefore, the coating of NPs is an excellent technique for enhancing their properties through electrostatic stabilization among particles and can decrease their agglomeration [30]. The surface coating of NPs is accomplished with a variety of materials, including bioactive compounds, organic and inorganic compounds, as well as natural and synthetic polymers [31]. HA is a natural organic matter containing alkyl and aromatic units attached with carboxylic acid, phenolic hydroxyl and quinone functional groups that can form complexes with iron and iron oxide [32]. HA is referred to as agricultural black gold, which is a prolific constituent of humic substances and is distributed in the environment. HA contains a high concentration of trace minerals and aids plant growth and is responsible for the structure and physio-chemical properties of soil. Humic substances in humin can be fractionated into fulvic acids, brown HA and grey HA based on their solubility at different pH levels. HA has a similar characteristic to Fe_3_O_4_ NPs, including improved dispersion stability for NPs [33] and electrostatic stabilization over a wide range of pH [34]. The synthesis of iron oxide is the most important parameter in producing crystalline polymorphs with desirable and distinctive properties that make them suitable for technological applications. The synthesis of iron oxide NPs can be achieved physically [35], chemically [36] and biologically [37]. Physical methods are easy to perform, but controlling the particle size is difficult. While in wet chemical preparation, the particle size can be somewhat controlled by adjusting the conditions. Therefore, chemical-based synthesis methods are mostly adopted due to their appropriateness, low production cost and high yield [38].

The present study aimed to: (1) prepare HA-coated Fe_3_O_4_ (Fe_3_O_4_/HA) NPs using a chemical co-precipitation method; (2) investigate the potential of Fe_3_O_4_/HA NPs in controlling the acremonium wilt disease of sorghum; (3) study its effect on growth and yield of Sorghum plants; and (4) investigate its potential toxicity on rats using chemical and histopathological investigations.

## 2. Materials and Methods

### 2.1. Preparations and Characterization of Coated Fe_3_O_4_/HA

HA coated Fe_3_O_4_ NPs (Fe_3_O_4_/HA) was prepared in the Department of Corn Diseases and Sugar Crops, Agricultural Research Center (ARC), Egypt, through a chemical co-precipitation method [39]. The size and morphology of NPs were determined by Transmission electron microscopy (TEM, JEOL JEM 1400) at National Research Center (NRC), Cairo, Egypt. Imaging of surface morphology and structure was conducted using Scanning Electron Microscopy (SEM, Quanta FEG250). The presence of synthesized Fe_3_O_4_/HA NPs was confirmed with Energy Dispersive X-ray Spectrometer (EDX) [40,41] at NRC, Cairo, Egypt. Size distribution, zeta potential and poly disparity index (Pdi) of NPs in suspension were determined using DLS by Zeta sizer (Nano ZS, Malvern, UK) at Regional Center for Food and Feed. ARC, Egypt.

### 2.2. Source and Inoculum Preparation of A. strictum

*A. strictum* used in this study was previously isolated from sorghum plants showing acremonium wilt symptoms. The identity of *A. strictum* was confirmed based on morphological characteristics. The inoculum was prepared by culturing *A. strictum* on PDA medium for 6 days at 30 °C.

### 2.3. Effect of Fe_3_O_4_/HA Nanoparticles on the Mycelial Growth of A. strictum

Effect of Fe_3_O_4_/HA NPs on the mycelial growth of *A. strictum* was determined on the PDA medium supplemented with yeast extract using the Poison media technique [42]. The PDA medium supplemented with different concentrations (10, 40 and 80 mg L^−1^) of Fe_3_O_4_/HA NPs was poured into Petri dishes. After solidification, the plates were inoculated at the center with fresh mycelial agar plugs of *A. strictum* and incubated at 28 ± 2 °C for 6 days. PDA plates without supplementing Fe_3_O_4_/HA NPs inoculated with mycelial plugs of *A. strictum* were run as controls. The diameter of mycelial growth was measured in millimeters (mm) and inhibition ratio was calculated.

### 2.4. Reaction of Twenty-One Genotypes of Grain Sorghum to A. strictum under Field Conditions

Twenty-one genotypes of grain sorghums were screened for their reaction to *A. strictum* under field conditions during the growing season of 2019. The trials were carried out at Agricultural Research Center (ARC) Farm, Giza, Egypt. The experimental plot was comprised of two rows of 4 m each, 60 cm apart and 20 cm between hills. After three weeks of sowing, sorghum plants were thinned to two plants per hill. Fourteen days post-flowering, artificial inoculation was performed, approximately 10 cm above the ground, by piercing a sterile needle followed by inserting a toothpick containing *A. strictum* mycelia into plant stalk between the leaf sheath and the stalk [43]. Four weeks later, the stalks were longitudinally sectioned, and the disease ranting was determined using the scale (Table 1). The infection levels, ranging from highly resistant to highly susceptible, were inferred according to the set criterion (Table 2) [44]. The experimental block design was completely randomized with three replicates for each genotype.

### 2.5. Effect of Fe_3_O_4_/HA Nanoparticels on A. strictum under Field Conditions

Fe_3_O_4_/HA NPs were evaluated for controlling *A. strictum* under field conditions during the seasons 2020 and 2021. Out of 21 genotypes, five sorghum genotypes, widely cultivated in Egypt, were selected after screening for their reaction to acremonium wilt disease in the first growing season (2019). A comparable experimental layout to the evaluation of genotypes was used. Grain sorghums were soaked in different concentrations (10, 40 and 80 mg L^−1^) of Fe_3_O_4_/HA NPs for 24 h prior to sowing. Additionally, the foliar spray of Fe_3_O_4_/HA NPs was performed twice on the plants with the respective NP concentrations; the first spray was performed one week before inoculation and the second spray was performed one week after the inoculation. The control plants were sprayed with sterile water. Four weeks after inoculation, disease severity was determined using the rating scale (Table 1). The experimental design block was completely randomized using strip split plot arrangement with three replicates. Treatments with Fe_3_O_4_/HA NPs (10, 40 and 80 mg L^−1^) and the control were horizontally allocated. While the genotypes Giza 113, LG 1, LG 3, Giza 3 and local 119 were vertically arranged in sub-plots. These five selected genotypes represent four categories as follows: LG 1 was highly susceptible; LG 3 was moderately susceptible; Giza 113 and Giza 3 were sorted as susceptible, while Local 119 was considered as resistant.

### 2.6. Gibberellic Acid Assay

The effect of Fe_3_O_4_/HA NPs on gibberellic acid levels in the five genotypes was determined, as previously described [45]. Samples were collected after 75 days of sowing and gibberellic acid was quantified using gas chromatography–mass spectrometry (GC-MS; Agilent 7000 Triple Quad) at the Regional Center for Food and Feed, ARC, Giza, Egypt.

### 2.7. Assessment of Growth and Yield Parameters

Yield parameters including a weight of 1000 grains, grain yield per plant and plant heights were determined at harvest time for all plants in each plot.

### 2.8. Enzyme Assays

Enzymatic activities of peroxidase and catalase were determined in leaves of sorghum plants after 75 days of sowing. Samples were collected and extracted with Tris-HCl buffer and the extracts were centrifuged at 10,000 rpm for 15 min at 4 °C. The resulting supernatants were separated as enzyme extracts. The peroxidase activity was determined [46] and expressed as ΔA 422 nm min^−1^ g^−1^ FW. The catalase activity was determined [47] and expressed as ΔA 240 nm min^−1^ g^−1^ FW.

### 2.9. Potential Toxicity of Fe_3_O_4_/HA NPs on Rats

Twelve male albino rats (average weight 120–150 gm) were used to assess the potential toxicity of Fe_3_O_4_/HA NPs. Rats obtained from Lab Animal House, Faculty of Veterinary Medicine, Benha University, Egypt. They were housed in well-ventilated stainless-steel wire cages under laboratory conditions at temperature of 18–24 °C and 12 h light and darkness. The experimental protocol was developed in accordance to ethical guidelines (BUFVTM 07-04-22). Rats were acclimatized for 5 days before the treatments. Rats were randomly divided into four groups; three groups (3 rats each) were caged individually and were administered grain sorghums treated with different concentrations (10, 40 and 80 mg L^−1^) of Fe_3_O_4_/HA NPs. The fourth group rats were fed on untreated grain sorghums and served as the control group.

After 21, 36 and 51 days of feeding, blood samples of rats were collected and subjected to CBC tests; including RBCs count [48], Hb and white blood cells (WBCs) [49].

Blood serum was used to assess the activity of serum AST and ALT as a marker of liver function [50] and creatinine level as a marker of kidney function [51].

### 2.10. Histopathological Examination

After 51 days of feeding on Fe_3_O_4_/HA NPs, autopsy samples were taken from the liver and kidneys of rats in different groups and fixed in 10% formalin saline for 24 h. The samples were thoroughly washed with water before being dehydrated with serial alcohol (methyl, ethyl and absolute ethyl) dilutions. Specimens were cleared in xylene and embedded in paraffin for 24 h at 56 °C in a hot air oven. Paraffin bees wax tissue blocks were prepared for sectioning at 4 microns thickness (Rotary microtome, Leitz, Germany). Tissue sections were obtained on glass slides, deparaffinized and stained with hematoxylin & eosin (H&E) staining and finally examined under a light electric microscope [52].

### 2.11. Data Analysis

Data were subjected for analysis of variance using software DSAASTAT Version 1.1 [53]. Differences between the means were inferred by Fisher’s protected least significant difference (LSD) at a level of 5% significance.

## 3. Results

### 3.1. Morphology and Size Distribution of Fe_3_O_4_/HA NPs

The TEM and SEM images confirmed that the Fe_3_O_4_/HA NPs were quasi-spherical in shape and had nearly uniform distribution, with particle sizes ranging from 60 to 72 nm (Figure 1A,C). The dynamic light scattering system (DLS) results revealed that the Fe_3_O_4_/HA NPs were heterogeneously distributed, with an average of approximately 174 nm representing 96% of the colloidal solution and a poly disparity index (Pdi) of 0.379 (Figure 1B). Zeta potential of Fe_3_O_4_/HA NPs exhibited a negative charge at −19.0 mv. The peaks of EDX results revealed the presence of iron, oxygen, carbon and other associated elements in the incorporated table (Figure 1E), therefore, confirming the synthesis of Fe_3_O_4_/HA NPs.

### 3.2. Inhibitory Effect of Fe_3_O_4_/HA NPs on the Mycelial Growth of A. striticum

Three used concentrations, 10, 40, and 80 mg L^−1^, of Fe_3_O_4_/HA NPs inhibited the mycelial growth of *A*. *striticum* in comparison to the controls (Figure 2A). The highest inhibition rate was observed at 80 mg L^−1^ concentration, with value of 62.22%, followed by 40 mg L^−1^ concentration, with a value of 44.44% (Figure 2B). The least inhibition rate 16.67% was observed at a 10 mg L^−1^ concentration (Figure 2B).

### 3.3. Reaction of Grain Sorghum Genotypes to Acremonium Wilt Disease

All tested twenty-one sorghum genotypes responded differently to *A. striticum* infection. The infected plants developed reddish to dark brown vascular bundles and stalks after 90 days of sowing. The disease rating was quantified based on their response to wilt disease and the disease severity index (DSI) and infection percent were calculated. All the genotypes were categorized into six groups, ranging from highly susceptible to highly resistant genotypes. The first group consisted of four highly resistant genotypes: LG 13, Local 119, Dorado, and LG 23. The second group was of moderately resistant genotypes and included Line c, H 301, local 129, and ICSR 93002 genotypes. The third group was the largest group and comprised of nine resistant genotypes: Assuit 14, Local 245, Sel 1007, LG 35, LG 47, H sh 1, H 304, H305, and ICSR 92003. Only one genotype (LG 3) was found to be moderately susceptible in the fourth group. Giza 3 and Giza 113 were two susceptible genotypes in the fifth group. Only one genotype, LG 1, constituted the sixth group and was highly susceptible (Table 3).

### 3.4. Effect of Fe_3_O_4_/HA NPs on Controlling Acremonium Wilt under Field Conditions

The effects of synthesized NPs on the severity of acremonium wilt on five sorghum genotypes were studied over two consecutive seasons (Table 4). The DR of the five tested genotypes was comparable in the two seasons. Nonetheless, the DR differed significantly (*p* ≤ 0.05) at the different concentrations of Fe_3_O_4_/HA NPs. The highest DSIs were observed at 10 mg L^−1^ concentration, and a concentration of 80 mg L^−1^ yielded the best control efficacy. Local 119, a highly resistant genotype, demonstrated the highest efficacy of 80% against *A. striticum*. Moreover, the concentrations 40 and 80 mg L^−1^ showed a great efficacy in controlling acremonium wilt on LG 1 (highly susceptible) and Giza 3 (susceptible) genotypes during the two seasons.

### 3.5. Effect of Fe_3_O_4_/HA NPs on the Growth and Yield of Grain Sorghum

Plant heights, grain weight and yield differed significantly (*p* < 0.05) among five genotypes in response to different concentrations of Fe_3_O_4_/HA NPs (Table 5, Table 6 and Table 7). The concentration of 40 mg L^−1^ was the most effective in increasing plant height, as compared to the other concentrations, with the highest plant height recorded in the two seasons being 390.33 and 392.33 cm for the Local 119 genotype, respectively (Table 5). The highest mean value of plant height for all the five genotypes was 382.67 cm in season 2020 and 377.27 cm in the season 2021.

The genotypes Giza 113 and Local 119 treated with 40 mg L^−1^ exhibited a substantial increase in the weight of 1000 grains. The highest mean value of 1000 grain weight treated with the concentration of 40 mg L^−1^ was 42.31 g and 42.73 g in the seasons 2020 and 2021, respectively (Table 6). Furthermore, the concentration of 40 mg L^−1^ had the greatest impact on the grain yield/plant for all the genotypes, with Giza 113 and Local 119 scoring the highest, with 84.10 g and 83.96 g for the season 2020 and 83.25 and 85.84 g for the season 2021. The LSD test (*p* < 0.05), however, revealed a low significant difference in the mean values of treatments and genotypes in terms of yield/plant. Conversely, no significant difference was observed in their interaction. The highest concentration, 80 mg L^−1^, had a distinct negative impact on the plant heights, 1000-grain weight and yield attributes (Table 5, Table 6 and Table 7).

### 3.6. Effect of Fe_3_O_4_/HA NPs on the Level of Gibberillic Acid in Grain Sorghum Genotypes

A concentration of 40 mg L^−1^ of Fe_3_O_4_/HA NPs led to a significant (*p* < 0.05) increase in the level of gibberellic acid (GA3) in five sorghum genotypes. In contrast, the concentration of 80 mg L^−1^ significantly (*p* < 0.05) downregulated the level of GA3 in all the genotypes when compared to the control and the lowest concentration of 10 mg L^−1^. GA3 levels remained unaffected by the lowest concentration, 10 mg L^−1^, and were comparable to untreated control plants (Figure 3).

### 3.7. Effect of Fe_3_O_4_/HA NPs on the Activity of Peroxidase and Catalase Enzymes in Grain Sorghum

The five sorghum genotypes treated with either 40 mg L^−1^ or 80 mg L^−1^ concentration of Fe_3_O_4_/HA NPs showed a significant (*p* < 0.05) increase in the enzymatic activity of peroxidase and catalase in healthy (uninoculated) plants in comparison with the infected plants (Figure 4A and Figure 5A). Notably, no significant increase in the activities of both enzymes were noted in infected sorghum genotypes.

The peroxidase activities of two genotypes, LG 3 and Giza 3, were comparable to each other in response to different concentrations of Fe_3_O_4_/HA NPs (Figure 4B). The peroxidase activity of all the five genotypes decreased substantially at concentrations of 40 and 80 mg L^−1^ in infected plants as compared to healthy plants (Figure 4B). Nonetheless, a variable trend was observed at 10 mg L^−1^ and in control plants. Two genotypes, Giza 3 and Local 119, showed a significant (*p* < 0.05) increase in peroxidase activity at a concentration of 10 mg L^−1^ and in control plants (Figure 4B).

Catalase activity was significantly reduced in all infected sorghum genotypes treated with 40 and 80 mg L^−1^ NP concentrations (Figure 5B). On the other hand, the catalase activity enhanced in infected sorghum plants treated with 10 mg L^−1^ as compared to healthy (uninfected) control plants (Figure 5B). Nonetheless, a highest increase in the catalase activity was noted in the Giza 3 and Local 119 genotypes (Figure 5B).

### 3.8. Potential Toxicity of Fe_3_O_4_/HA NPs in Rats

Hematological examinations of rats fed with grain sorghums treated with different concentrations of Fe_3_O_4_/HA NPs did not show any significant difference until 36 days of feeding. Likewise, no significant increase in the red blood cells (RBCs) count was observed at 36 days of post-feeding, but the hemoglobin (Hb) level significantly increased (*p* < 0.05) at 36 days and continued to increase at 51 days at 40 and 80 mg L^−1^ concentrations. However, at 51 days of post feeding, a significant increase in the RBCs count was observed at 40 and 80 mg L^−1^ concentrations (Table 8).

The enzymatic activity of alanine aminotransferase (ALT) and aspartate aminotransferase (AST) was used to assess the liver function of rats feeding on grain sorghums treated with different concentrations of Fe_3_O_4_/HA NPs (Table 9). From 21 to 51 days of feeding, rats fed on 80 mg L^−1^ concentration experienced a gradual increase in ALT levels, whereas the level of AST was not highly affected as compared to the control.

The level of creatinine remained unaffected by all concentrations when compared to the control until 36 days of feeding. However, after 51 days, there was a marked increase in the creatinine level in the rats feeding on 80 mg L^−1^ concentration of Fe_3_O_4_/HA NPs which recorded 0.88 mg dL^−1^, but this did not indicate kidney failure (Table 9).

#### Histopathological Examination

No morphological changes were observed in the hepatic parenchyma of rats administered with untreated grain sorghums or grain sorghums treated with 10 mg L^−1^ of Fe_3_O_4_/HA NPs (Figure 6A,B). The kidneys of these rats showed no histopathological changes in the glomeruli and tubules at the cortex (Figure 7A,B), whereas the hepatic parenchyma of rats administered with 40 mg L^−1^ of Fe_3_O_4_/HA NP-treated grain sorghums exhibited mild pathological changes in the portal areas, including inflammatory cells infiltrations and portal vein dilatations (Figure 6C). The kidneys of these rats showed a few focal inflammatory cells infiltration between the sclerotic dilated blood vessels, glomeruli and tubules (Figure 7C). The hepatic parenchyma of rats administered with grain sorghums treated with 80 mg L^−1^ of Fe_3_O_4_/HA NPs showed several pathological alterations, such as massive inflammatory cells’ infiltration between the fatty changed hepatocytes all over the parenchyma (Figure 6D). Furthermore, histopathological alterations, such as focal massive inflammatory cell aggregation in between the degenerated necrosis tubules and vacuolization in the endothelial cells lining the glomeruli, were clearly observed in the kidneys of these rats (Figure 7D).

## 4. Discussion

The current study was the first to investigate the inhibitory effect of Fe_3_O_4_/HA NPs against a fungal pathogen, A. striticum. The TEM and SEM images confirmed the spherical shape of Fe3O4/HA [34], with sizes ranged from 60–72 nm. However, the hydrodynamic size obtained through DLS (Figure 1B) was larger (174 nm) than the size obtained through TEM. These findings corroborate an earlier report wherein the different sizes of iron oxide NPs were observed [54]. The average size distribution of Fe_3_O_4_/HA NPs in colloids was 174 nm, representing 96% of all colloidal solution, which is in agreement with the previous findings [32] and contrasted to another where size of NPs were 239 nm [19]. Iron oxide NPs usually range from 1 to 100 nm in size, but the higher size observed here may likely be due to the HA coating. Moreover, the Pdi recorded 0.379, indicating moderate polydispersity which is in close proximity with previously recorded 0.423 Pdi [55]. The zeta potential was −19.0 mv, indicating that Fe_3_O_4_/HA NPs have a hydrodynamic diameter in the nanoscale which is similar to previously reported [40].

Different concentrations 10, 40 and 80 mg L^−1^ of Fe_3_O_4_/HA NPs significantly (*p* < 0.05) reduced the mycelial growth of *A. striticum* over the control. Our finding is supported by earlier studies that exhibited the antimicrobial activity of Fe_3_O_4_ and HA against fungal and bacterial agents [56,57,58,59,60,61]. Iron oxide NPs have not only been shown to boost the plant growth but also exhibit the antifungal activities against a variety of fungal pathogens [58]. In the poisoned food technique, iron NPs inhibited the mycelial growth of *Alternaria alternata* up to 87.9% at a 1.0 mg mL^−1^ concentration [62]. Iron oxide NPs exhibited significant antimycotic activity against *Aspergillus niger* and *Mucor piriformis*, but were more active against *Mucor piriformis* [63], inhibited *Trichothecium roseum* spore germination (up to 87.74%) and *Cladosporium. herbarum* (up to 84.89%) [56]. The foliar spray and soil application of iron NPs improved the morpho-physiochemical attributes, morphological traits (sprouting percentage, number of leaves, plant biomass, root attributes), and biochemical attributes (such as chlorophyll, sugar content, and enzymatic activity of antioxidants) [54]. HA has been shown to have antimicrobial activity against a variety of pathogenic bacteria [64] and fungi, including *Botrytis cinerea*, *Fusarium graminearum*, *Physalospora piricola*, *Phytophthora infestans*, *Rhizoctonia cerealis*, and *Sclerotinia sclerotiorum* [65,66]. The exact mechanisms underlying the antifungal activity of Fe_3_O_4_/HA NPs need to be investigated and could be the subject of futuristic studies. However, antifungal activity could be accredited to either iron oxide or HA or could be a combined effect of these two. Previous studies have bounded the antimicrobial activity of iron oxide NPs to their extremely tiny size and larger surface area, which reduces oxygen supply for respiration [67] or causes cell damage due to the oxidative stress generated by reactive oxygen species, ultimately leading to the inhibition of microbial growth [57,68,69,70]. Furthermore, iron reacts with oxygen to create hydrogen peroxide, which then reacts with ferrous ions to form hydroxyl radicals that disrupt the cell components [71]. On the other side, HA controls plant diseases by inducing systemic acquired resistance [72], or by inhibiting the growth of microorganisms due to its unique chemical composition and functional group (COOH) properties [73].

The screened sorghum genotypes responded variably to acremonium wilt. At present, few works have been performed on the screening of sorghum genotypes against acremonium wilt under Egyptian conditions. Available studies in Egypt reported that the hybrids H-301 and H-302 were resistant to acremonium wilt disease, and stated that the genotype Giza 15 (Egyptian) was susceptible and Dorado (USA) was resistant [44,74]. Fortunately, this work provides an advance for the wide screening of for 21 sorghum genotypes, and thus represents a useful datum for researchers in the future.

The phenotypic traits of sorghum genotypes showed dramatic effects on the resulting plants’ growth and yield upon exposure to the Fe_3_O_4_/HA treatments. The genotypes treated with 40 mg L^−1^ revealed longer shoots and higher yield. No significant changes were observed in the growth and yield of sorghum genotypes treated either with 10 mg L^−1^ or untreated control. It is well known that iron is a very important trace element for photosynthetic processes and plant growth. Many investigators pointed out the significant effect of iron on the growth, yield and other biochemical parameters of many plants [75,76]. Along with this, Li et al. demonstrated that foliar spraying with suitable concentrations (20–50 mg L^−1^) of iron oxide NPs could significantly promote the growth of *P. heterophylla* and increase the root yield per unit area under field conditions [77]. In fact, the higher concentration 80 mg L^−1^ caused a negative effect on the growth of all genotypes. Other studies reported similar results where they found out that the high concentrations of Fe_3_O_4_ NPs had an inhibitory effect on growth due to accumulation in some cells, leading to stress effects [77,78]. The importance of HA cannot be ignored as many studies explored its beneficial application as biological amendments for maintaining plant growth [79,80,81,82].

Several studies have shown that NP applications increased the activity of different antioxidant enzymes such as catalase, superoxide dismutase and peroxidase [82,83,84]. As per our findings, the stimulating effect of Fe_3_O_4_/HA NPs was directly proportional to their tested concentrations. The higher concentrations of 40 and 80 mg L^−1^ increased the activity of peroxidase and catalase in non-infected plants compared to the infected one. Despite stimulating the activity of oxidative enzymes, the higher concentration of 80 mg L^−1^ negatively affected the growth of sorghum genotypes. However, Haydar et al. noted that the activities of all studied enzymes except peroxidase increased along with the plant growth [54]. The dissimilarities in published results might be due to the difference in NP size, culture condition or plant species used in the studies [77].

The current study’s goal was to assess the potential toxicity of Fe_3_O_4_/HA NPs in male albino rats via assessing any changes in complete blood count (CBC), liver and kidney functions and their histological features. Biocompatibility and the potential impact of nanomaterials and polymers on the immune system of model animals and on immune-related hematological parameters have been explored [85,86,87]. The results of the current study indicated that the administration of lower concentrations 10 mg L^−1^ was not associated with histopathological changes neither in the liver nor in the kidney and no significant changes in liver enzymes were recorded. This is consistent with the findings that lower doses of Fe_3_O_4_ had no negative effects on liver tissues and enzymes [86,88]. Moreover, the magnetic NPs of 50 nm size did not cause apparent toxicity under the experimental conditions [89]. Compared to unexposed control, the tested concentrations of chitosan-coated iron oxide NPs had no effect on the creatinine level in rats and did not create any toxicity in the kidney [90]. These findings confirm that lower doses have good biocompatibility in mice. However, in this study, administration to a higher dose 40 mg L^−1^ revealed mild histopathological effects on liver and kidney. The higher dose 80 mg L^−1^-induced histopathological alterations in the kidney and liver as well as increased the level of ALT and creatinine after 51 days of feeding. This increment might be due to AST and ALT leakage from the liver cytosol to blood flow and led to an increase in their activities [91,92]. Thus, AST and ALT enzyme are markers for liver damage, confirmed by our histopathological findings (fatty changed hepatocytes all over the parenchyma). Thus, the increase in the ALT enzyme shows the damaging effect of iron oxide NPs on liver cells. These finding are supported with those reported earlier. The possible reasons that can explain these conflicting reports include the physicochemical properties of the NPs, such as shape, size and solubility. Additionally, experimental conditions, exposure time and doses can cause variation in results [93]. 

## 5. Conclusions and Further Perspectives

The use of nanomaterials in agriculture, especially nanofertilizers having antimycotic properties, could pave the way for smart agriculture. The current study significantly adds new information and yielded promising results about the antimycotic and biofertilizer effects of Fe_3_O_4_/HA NPs. It was shown that the concentrations of 40 and 80 mg L^−1^ had a great efficacy in controlling the causal agent of acremonium wilt in vivo and in vitro. Unfortunately, the higher concentrations had a negative impact on the activities of peroxidase and catalase enzymes in infected sorghum plants. Only a concentration of 40 mg L^−1^ was the most effective in increasing the gibberellic acid level as well as the growth of grain sorghum plants. Conclusively, Fe_3_O_4_/HA NPs at a concentration of 40 mg L^−1^ could be regarded as safe and could potentially be opted in agriculture applications against the filamentous fungus *A*. *strictum*. However, the molecular mechanisms underlying Fe_3_O_4_/HA NPs-mediated antimycotic mechanisms needs further investigation and could be a subject of futuristic studies. Furthermore, the application of iron oxide NPs in agriculture reveals one more issue: utilizing Fe_3_O_4_/HA NPs showed some limitations at a higher dose of 80 mg L^−1^. Therefore, the application of Fe_3_O_4_/HA NPs still requires more effort to achieve the hope for the future. These efforts might focus on the use of alternative coating materials and stabilizing the surface of iron oxide NPs in order to reduce toxicity and increase biocompatibility.

## Figures and Tables

**Figure 1 polymers-14-03099-f001:**
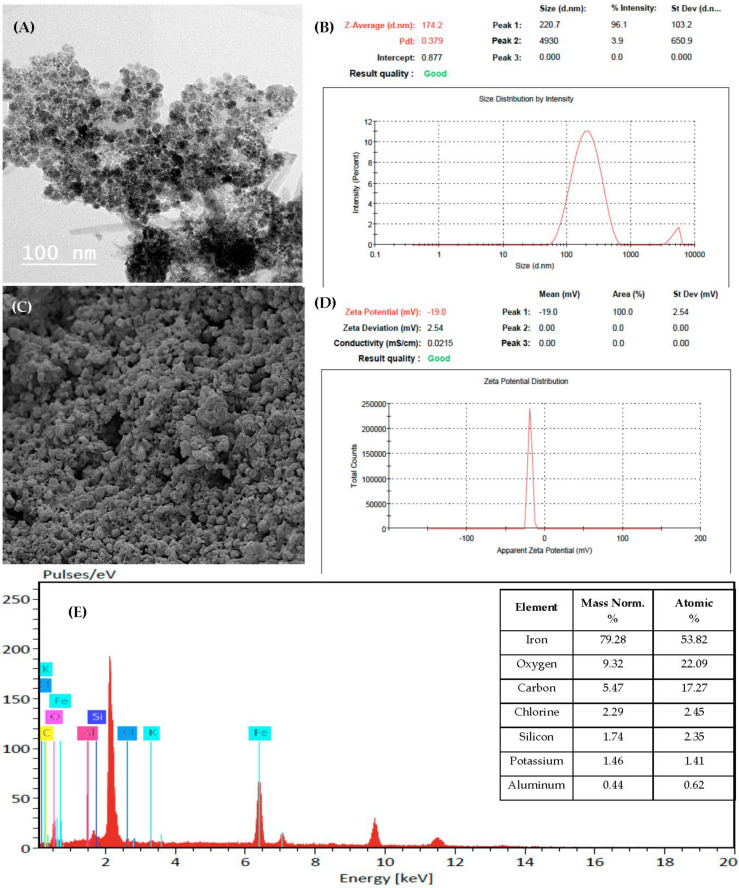
TEM image (**A**), Zeta average diameter (**B**), SEM image (**C**), Zeta potential (**D**) and EDX profile of Fe_3_O_4_/HA NPs (**E**).

**Figure 2 polymers-14-03099-f002:**
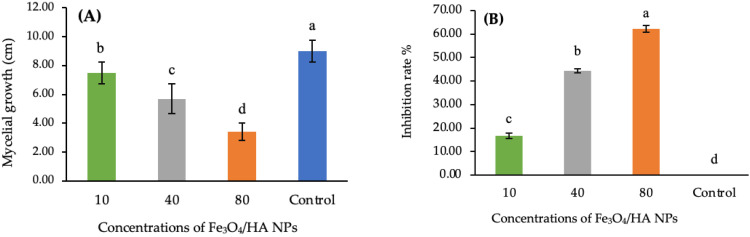
Inhibitory effect of Fe_3_O_4_/HA NPs on the mycelial growth of *A*. *striticum* (**A**) and inhibition rate (**B**). Same letter in the figure means no significant differences between the values.

**Figure 3 polymers-14-03099-f003:**
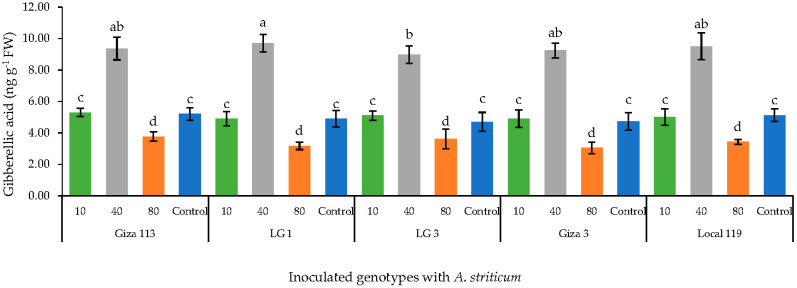
Effect of Fe_3_O_4_/HA NPs on the level of gibberellic acid (GA3) in grain sorghum genotypes. Same letter in the figure means no significant differences between the values.

**Figure 4 polymers-14-03099-f004:**
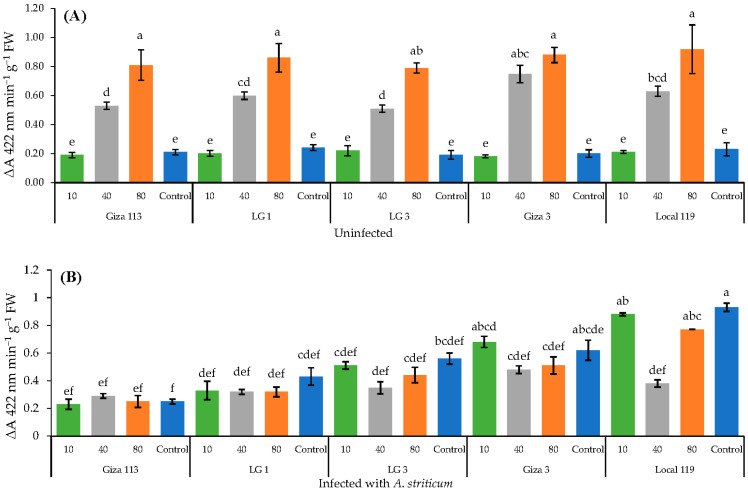
Activity of peroxidase in grain sorghum plants uninfected (**A**) and infected with *A. striticum* (**B**) in response to application of Fe_3_O_4_/HA NPs. Same letter in the figure means no significant differences between the values.

**Figure 5 polymers-14-03099-f005:**
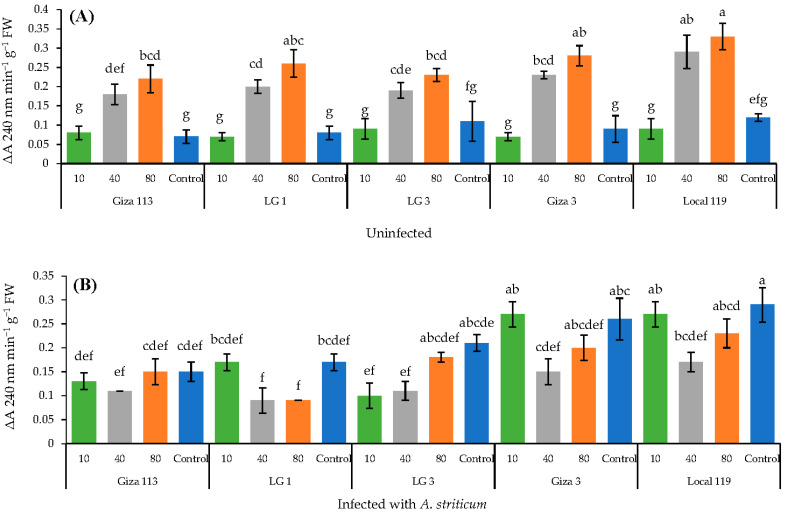
Activity of catalase in grain sorghum plants uninfected (**A**) and infected with *A. striticum* (**B**) in response to application of Fe_3_O_4_/HA NPs. Same letter in the figure means no significant differences between the values.

**Figure 6 polymers-14-03099-f006:**
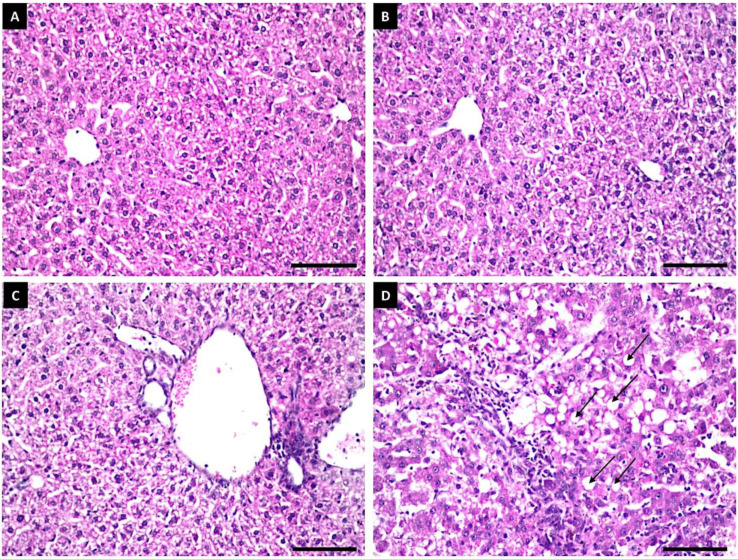
Histopathological changes in the livers of male albino rats fed with grain sorghums treated with Fe_3_O_4_/HA NPs; no histopathological changes were recorded in the hepatocytes and central vein of control (untreated) and treated with 10 mg L^−1^ rats (**A**,**B**); few inflammatory cells infiltration and dilatation in the portal vein of rats exposed to 40 mg L^−1^ Fe_3_O_4_/HA NPs (**C**); massive inflammatory cells infiltration was detected in between the fatty changed hepatocytes all over the parenchyma (black arrow) of rats exposed to 80 mg L^−1^ Fe_3_O_4_/HA NPs (**D**). Scale bar represents 100 μm.

**Figure 7 polymers-14-03099-f007:**
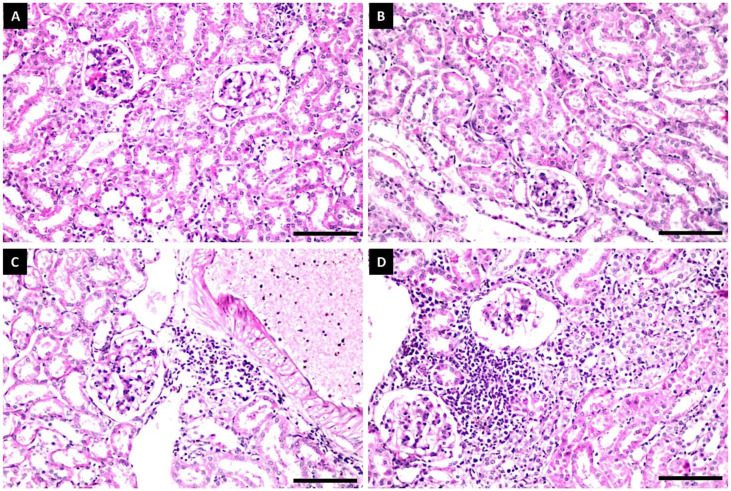
Histopathological changes in the kidney of male albino rats fed with grain sorghums treated with Fe_3_O_4_/HA NPs; no histopathological alteration in the glomeruli and tubules at the cortex were recorded in untreated and treated rats with 10 mg L^−1^ (**A**,**B**); few focal inflammatory cells infiltration was detected in between the sclerotic dilated blood vessels, glomeruli and tubules of rats exposed to 40 mg L^−1^ Fe_3_O_4_/HA NPs (**C**); focal massive inflammatory cell aggregation in between the degenerated necrosis tubules as well as vacuolization in the endothelial cells lining the glomeruli of rats exposed to 80 mg L^−1^ Fe_3_O_4_/HA NPs (**D**). Scale bar represents 100 μm.

**Table 1 polymers-14-03099-t001:** Disease rating scale adopted according to IDIN-instruction manual.

Numerical Grade	Degree of Infection
0.1	Minimal reaction, indistinguishable from that to a sterile toothpick.
0.2	Discoloration centered around the wound, progressing in the superficial parts of the stalk, but not reaching either node.
0.5	Extensive discoloration progressing in the central part of the stalk.
0.8	Discoloration reaching one or both nodes superficially or forming a cylinder.
1.0	Most or all of one internode discolored with no penetration of nodal areas.
1.1	Slight penetration of one or both nodes.
1.2	Nearly complete penetration of one or both nodes,
1.5	Penetration of one node and slight invasion of next internode.
2.0	More than one internode but not more than two affected; infection must have spread through at least one internode.
2.5	Penetration of two nodes and slight invasion of distal internode.
3.0	Infection passed through two or more internodes.
4.0	Extensive invasion of plant but not killed.
5.0	Death of plant due to stalk-rot.

**Table 2 polymers-14-03099-t002:** Category of *A. striticum* infestation based on numerical categorization and level of infection.

No.	Category Numerical	Level of Infection
1	0.0–0.5	Highly resistant
2	0.6–1.0	Resistant
3	1.1–1.5	Moderately resistant
4	1.6–3.0	Moderately susceptible
5	3.1–4.0	Susceptible
6	4.1–5.0	Highly Susceptible

**Table 3 polymers-14-03099-t003:** Reaction of twenty-one grain sorghum genotypes for acremonium wilt disease during season 2019.

No.	Genotypes	DR ± St.Dev.	Group Type
1	LG 13	0.17 ± 0.03	HR
2	Assuit 14	0.67 ± 0.10	R
3	Line c	1.11 ± 0.10	MR
4	Local 119	0.52 ± 0.08	HR
5	Local 245	0.72 ± 0.04	R
6	Dorado	0.39 ± 0.01	HR
7	LG 23	0.46 ± 0.07	HR
8	Sel 1007	0.83 ± 0.04	R
9	LG 35	0.89 ± 0.10	R
10	LG 47	0.73 ± 0.03	R
11	H sh 1	0.72 ± 0.05	R
12	H 301	1.25 ± 0.06	MR
13	H 305	0.90 ± 0.04	R
14	H 306	0.93 ± 0.02	R
15	LG 1	4.12 ± 0.11	HS
16	Giza 3	3.45 ± 0.22	S
17	Giza 113	3.06 ± 0.13	S
18	Local 129	1.29 ± 0.08	MR
19	ICSR 92003	1.02 ± 0.08	R
20	ICSR 93002	1.08 ±0.08	MR
21	LG 3	2.03 ± 0.06	MS
Average		1.25	
LSD 0.05		0.14	

Abbreviations used in the table are: the least significant difference (LSD), highly susceptible (HS), moderately susceptible (MS), susceptible (S), resistant (R), moderately resistant (MR), and highly resistant (HR), standard deviation (St.Dev.) and disease rating (DR).

**Table 4 polymers-14-03099-t004:** Effect of Fe_3_O_4_/HA NPs on acremonium wilt of grain sorghum during seasons 2020 and 2021.

Genotypes	Concentration (mg L^−1^)	Season 2020		Season 2021	
		DR ± St.Dev.	Efficacy (%)	DR ± St.Dev.	Efficacy (%)
Giza 113	10	3.2 ± 0.20	17.3	3.53 ± 0.15	7.8
	40	1.8 ± 0.10	53.49	1.87 ± 0.25	51.17
	80	1.53 ±0.06	60.47	1.5 ± 0.30	60.84
	Control	3.87 ± 0.15	-	3.83 ± 0.21	-
LG 1	10	3.3 ± 0.20	22.72	3.37 ± 0.31	22.25
	40	1.73 ± 0.15	59.48	1.83 ± 0.06	57.74
	80	1.2 ± 0.10	71.9	1.23 ± 0.06	71.59
	Control	4.27 ± 0.06	-	4.33 ± 0.25	-
LG 3	10	2.03 ± 0.06	2.4	2.5 ± 0.20	8.42
	40	1.3 ± 0.10	37.2	1.23 ±0.06	54.95
	80	0.97 ± 0.21	53.14	0.9 ± 0.17	67.03
	Control	2.07 ± 0.31	-	2.73 ± 0.31	-
Giza 3	10	2.93 ± 0.12	14.0	2.77 ± 0.21	20.17
	40	1.47 ± 0.06	58.82	1.4 ± 0.10	59.65
	80	1.1 ± 0.20	69.19	0.97 ± 0.12	72.04
	Control	3.57 ± 0.25	-	3.47 ± 0.15	-
Local 119	10	0.37 ± 0.06	26.0	0.4 ± 0.10	20.0
	40	0.13 ± 0.06	74.0	0.13 ± 0.06	74.0
	80	0.1 ± 0.00	80.0	0.1 ± 0.00	80.0
	Control	0.5 ± 0.10	-	0.5 ± 0.00	-
LSD of treatments		0.11		0.48	
LSD of genotypes		0.18		0.53	

Abbreviations used in the table are standard deviation (St.Dev.) and disease rating (DR).

**Table 5 polymers-14-03099-t005:** Effect of Fe_3_O_4_/HA NPs on plant height of grain sorghum during the seasons of 2020 and 2021.

	Plant Height (cm)											
Concentration (mg L^−1^)	Season 2020						Season 2021					
	Giza 113	LG 1	LG 3	Giza 3	Local119	Mean	Giza 113	LG 1	LG 3	Giza 3	Local119	Mean
10	370.33	363.00	355.33	375.00	376.67	368.07	371.67	362.67	353.50	373.44	375.67	367.39
40	386.00	377.00	371.33	388.67	390.33	382.67	382.67	370.00	360.00	381.33	392.33	377.27
80	311.00	307.33	297.33	318.67	325.67	312.00	309.37	300.14	291.33	315.47	324.67	308.20
Control	370.33	360.33	353.33	371.33	375.33	366.13	370.67	361.44	350.00	372.00	375.00	365.82
Mean	359.42	351.92	344.33	363.42	367.00	357.22	358.59	348.56	338.71	360.56	366.92	354.67
	F test			LSD at 0.05			F test			LSD at 0.05		
Treatments	**			3.17			**			3.57		
Genotypes	**			2.00			**			2.71		
Interaction				NS						NS		

** means significant 5% at level of probability.

**Table 6 polymers-14-03099-t006:** Effect of Fe_3_O_4_/HA NPs on the 1000-grain weight of grain sorghum during the seasons of 2020 and 2021.

	1000-Grain Weight (g)											
Concentration (mg L^−1^)	Season 2020						Season 2021					
	Giza 113	LG 1	LG 3	Giza 3	Local119	Mean	Giza 113	LG 1	LG 3	Giza 3	Local119	Mean
10	42.45	36.60	38.55	40.47	41.74	39.96	40.78	36.00	39.89	42.89	41.90	40.29
40	45.31	38.07	40.55	43.27	44.32	42.31	43.35	38.30	42.60	44.34	45.08	42.73
80	32.86	27.82	30.39	31.69	34.91	31.53	32.29	26.43	30.26	31.29	33.05	30.66
Control	42.22	36.20	38.27	40.26	41.50	39.69	40.50	35.73	39.03	42.75	41.34	39.87
Mean	40.71	34.67	36.94	38.92	40.62	38.37	39.23	34.12	37.95	40.32	40.34	38.39
	F test			LSD at 0.05			F test			LSD at 0.05		
Treatments	**			1.3			**			0.84		
Genotypes	**			1.08			**			0.6		
Interaction				NS						1.2		

** mean significant 5% at level of probability.

**Table 7 polymers-14-03099-t007:** Effect of Fe_3_O_4_/HA NPs on the grain yield/ plant for grain sorghum plants during the seasons of 2020 and 2021.

	Grain Yield/Plant (g)											
Concentration (mg L^−1^)	Season 2020						Season 2021					
	Giza 113	LG 1	LG 3	Giza 3	Local119	Mean	Giza 113	LG 1	LG 3	Giza 3	Local119	Mean
10	79.41	71.51	73.05	74.81	80.93	75.94	78.71	71.89	72.68	75.51	81.86	76.13
40	84.10	74.44	75.64	76.59	83.96	78.95	83.25	73.67	75.91	79.04	85.54	79.48
80	70.81	62.54	60.88	65.01	71.82	66.21	67.72	62.16	59.61	61.69	70.57	64.35
Control	79.09	71.30	73.17	74.18	80.29	75.60	78.36	71.67	72.49	75.16	81.48	75.83
Mean	78.35	69.95	70.68	72.65	79.25	74.18	77.01	69.85	70.17	72.85	79.86	73.95
	F test			LSD at 0.05			F test			LSD at 0.05		
Treatments	**			0.77			**			0.98		
Genotypes	**			1.11			**			0.62		
Interaction				NS						1.25		

** mean significant 5% at level of probability.

**Table 8 polymers-14-03099-t008:** Hematological attributes of male albino rats fed on grain sorghum treated with Fe_3_O_4_/HA NPs.

Days after Feeding	Concentration (mg L^−1^)	CBC Analysis		
		RBCs (10^6^)	WBCs (10^3^)	Hb (mg dL^−1^)
21	10	7.63	6.83	14. 2
	40	7.47	7.43	14. 1
	80	7.5	7.6	14.5
	Control	7.42	7.57	14.1
36	10	7.52	7.32	14.63
	40	7.72	7.83	14.78
	80	7.8	7.6	15
	Control	7.38	7.08	14.5
51	10	7.87	7.15	14.49
	40	8.29	7	15.3
	80	8.88	7.5	16.1
	Control	7.49	7.25	14.4
LSD 0.05 Treatment		0.54	0.52	0.4
LSD 0.05 Days of feed		0.47	0.45	0.35

Abbreviation used in the table are complete blood count (CBC), white blood cells (WBCs), red blood cells (RBCs), and hemoglobin (Hb).

**Table 9 polymers-14-03099-t009:** Liver and kidney functions analysis of male albino rats fed on grain sorghum treated with Fe_3_O_4_/HA NPs.

Days after Feeding	Concentration (mg L^−1^)	Blood Chemistry		
		AST (u L^−1^)	ALT (u L^−1^)	Creatinine (mg dL^−1^)
21	10	79	34.4	0.57
	40	83	36.7	0.57
	80	86	57.2	0.6
	Control	80	33.5	0.58
36	10	85	35	0.6
	40	84.6	38	0.59
	80	86	65.7	0.61
	Control	84	34.5	0.6
51	10	83.8	35	0.67
	40	84.3	40.7	0.69
	80	85	80.3	0.88
	Control	83	35	0.66
LSD 0.05 Treatment		0.88	0.88	0.07
LSD 0.05 Days of feed		0.76	0.76	0.06

## Data Availability

Not applicable.

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
