# Peer review of "Humic Acid-Coated Fe3O4 Nanoparticles Confer Resistance to Acremonium Wilt Disease and Improve Physiological and Morphological Attributes of Grain Sorghum"

_polymers, 2022, doi:10.3390/polym14153099_

Round 1

Reviewer 1 Report

This article does not look worthy and cannot be recommended for publication in this form and at least needs some revision.

1.     Introduction. More motivational explanations needed, why Fe3O4? Why Fe3O4 and not Fe2O3. If we are talking about nanomaterials, then more information about the sizes and preparation methods, as well as their possible influence on the results, is absolutely necessary. For a wider readership, I would like more information about the methods for obtaining Fe3O4. See examples, some of them published this year in MDPI journals.

Serga, V.et al . Impact of Gadolinium on the Structure and Magnetic Properties of Nanocrystalline Powders of Iron Oxides Produced by the Extraction-Pyrolytic Method. Materials 202013, 4147

Li, Y.; et al. Superparamagnetic α-Fe2O3/Fe3O4 Heterogeneous Nanoparticles with Enhanced Biocompatibility. Nanomaterials 202111, 834

Dukenbayev, K.; et al . Fe3O4 Nanoparticles for Complex Targeted Delivery and Boron Neutron Capture Therapy. Nanomaterials 20199, 494. https://doi.org/10.3390/nano9040494

Such short overview paragraph will arouse more interest among readers.

2.     Table 1. Accuracy in hundredths of % requires justification, because it seems excessive

3.     Table 2. The same question as above. Maybe show the error bar?

Reviewer 2 Report

Journal: Polymers (ISSN 2073-4360)

Manuscript ID: polymers-1810049

Type: Article

Title: Humic acid-coated Fe3O4 nanoparticles confer resistance to acremonium wilt disease and improve physiological and morphological attributes of grain Sorghum.

Authors: Sherif Mohamed El-Ganainy * , Amal M. El-Bakery , Heba M. Hafez , Ahmed Mahmoud Ismail * , Ali Zein El-Abdeen , Abed Abd Elgalel Ata , O A. Y. Abd Elraheem , Y. M. Y. El Kady , Ahlam Farouk Hamouda , Hossam S. El-Beltagi , Wael F. Shehata , Tarek A. Shalaby , Ahmed Osman Abbas , Mustafa I. Almaghasla , Muhammad N. Sattar , Zafar Iqbal.

a)           Abstract: Expand the abstract to include all the acquired findings.

b)          Why the author didn’t use SEM images?

c)           Why the author didn’t use EDX measurement in order to measure the composition of the samples?

d)          Why the author didn’t utilize XRD diffraction to indicate the forming of Fe2O3 and Fe3O4 phases?

e)           Why the author didn’t measure the optical properties of the nanoparticles used?

f)            For references, choose recent refs. Please, refer to these refs. EDS analysis

DOI: https://doi.org/10.1007/s11082-019-2015-5

DOI: https://doi.org/10.1007/s11082-016-0812-7

 Best Regards

Reviewer 3 Report

Manuscript ‘Humic acid-coated Fe3O4 nanoparticles confer resistance to acremonium wilt disease and improve physiological and morphological attributes of grain Sorghum

In this research aimed to assess the efficacy of humic acid (HA) coated Fe3O4 (Fe3O4/HA) nanoparticles in controlling acremonium wilt disease and improving sorghum growth and yields. During the season 2019, twenty-one sorghum genotypes were screened to assess their response to Acremonium striticum via artificial infection under field conditions and each genotype was assigned to one of six groups, ranging from highly susceptible to highly resistant.

            This research is interesting from the point of view of antibacterial materials, therefore, it well suitable for publishing and can be published after some minor improvements:

Introduction is very brief and limited, therefore, this part of the manuscript could be extended and advanced; e.g. authors are mentioning that ‘Different NPs have successfully been used to control different bacterial and viral pathogens’, this part of overview can be extended by referring antibacterial/antiviral activity of classical materials (Comparative study of Antifungal Activity of Silver and Gold Nanoparticles Synthesized by Facile Chemical Approach. Journal of Environmental Chemical Engineering 2018. 6, 5837-5844. // Antibacterial and antifungal activity of silver nanospheres synthesized by tri-sodium citrate assisted chemical approach. Vacuum 2017, 146, 259-265.), which are used for antibacterial/antiviral various environments, however are more expensive in comparison of here proposed Fe3O4-based structures.

According to the most of Papers published in MDPI Polymers, it is reasonable to shift section of ‘Materials and Methods’ before ‘Results’ section to make article structurally more similar to others.

‘Potential toxicity of Fe2O3/HA NPs in rats’ could be additionally compared/discussed with a toxicity/biocompatibility studies of some other polymer based particles (Biocompatibility of polypyrrole particles: an in vivo study in mice. Journal of Pharmacy and Pharmacology 2007, 59, 311–315.) studied by assessment at some extent similar haematological characteristics of rodent animals threated by these polymeric particles.

Conclusions are very brief and limited, therefore, some extension would be preferred. In addition to conclusions some further perspectives in the application/development of NPs for antibacterial/antiviral purposes could be discussed.

Round 2

Reviewer 1 Report

The authors have successfully improved the manuscript, which can now be recommended for publication